# Stochastic Ordering of Stationary Distributions of Linear Recurrences: Further Results and Economic Applications

Christian Di Pietro [1,†], Mariafortuna Pietroluongo [1,‡] and Marco M. Sorge [2,*,†,§]

1   Dipartimento di Studi Aziendali ed Economici, University of Napoli Parthenope, 80133 Naples, Italy; christian.dipietro@uniparthenope.it (C.D.P.); mfortuna.pietroluongo@uniparthenope.it (M.P.)
2   Dipartimento di Scienze Economiche e Statistiche, University of Salerno, 84084 Fisciano, Italy
*   Correspondence: msorge@unisa.it
†   Current address: Centre for Economic and Labour Policy Evaluation (CELPE), 84084 Fisciano, Italy.
‡   Current address: Centro di Ricerca Interdipartimentale in Sviluppo Economico e Istituzioni (CRISEI), 80133 Naples, Italy.
§   Current address: Center for Studies in Economics and Finance (CSEF), 80126 Naples, Italy.

**Abstract:** We investigate pairwise stochastic comparisons of stationary solutions to the linear recurrence $X_{t+1} = A_t X_t + B_t$, where $A_t$ and $B_t$ are non-negative random variables. We establish novel order-preserving properties, which enable us to obtain comparison theorems about well-known measures of conditional size, tail variability and skewness across probability distributions. While useful in studies of ergodic wealth accumulation processes and the persistence of inequality, our results can fruitfully be exploited to conduct comparative statics exercises in structural models entailing Kesten-type reduced-form representations. An application of our analysis to a dynamic asset accumulation model uncovers the qualitatively similar effects of capital income and earnings taxation on expected wealth concentration over higher quantiles as well as on conditional upper tail dispersion of wealth holdings, qualifying previous results that solely rely on the determination of Pareto exponents.

**Keywords:** stochastic orders; linear recurrences; wealth distribution; Kesten equations

## 1. Introduction

There exists by now extensive literature in applied mathematics and probability theory dealing with *stochastic recurrence equations* of the form

$$X_{t+1} = A_t X_t + B_t, \quad t \in \mathbb{Z}^+, \tag{1}$$

where $((A_t, B_t))_{t \in \mathbb{Z}^+}$ are pairs of continuous, real-valued, i.i.d. non-negative random variables. While early studies mostly focused on issues of existence and uniqueness of strictly stationary solutions $X^s$ to (1) and properties of the underlying limit support, particular emphasis has recently been placed on the characterization of the tail behavior of the marginal distributions of such solutions, a main result being that, under mild regularity conditions on the distribution of inputs $(A, B)$, the tails of the output $X^s$ are asymptotic to a power law, see, e.g., Kesten (1973), Goldie (1991) and Buraczewski et al. (2016) for details.[1]

In this paper, we derive novel order-preserving properties in the context of linear random recurrences, which enable us to obtain pairwise comparison theorems about well-known measures of *conditional size*, *tail variability* and *skewness* across stationary solutions to (1). Although a full grasp of our results requires introducing the reader to basic notions of stochastic ordering of probability distributions, we can summarize them in a non-technical

fashion as follows: consider two alternative input pairs $(A, B)$ and $(\hat{A}, \hat{B})$ for the linear model (1), and let $X^s$ and $\hat{X}^s$ be the corresponding stationary solutions to (1), provided they exist, then:

1. *Size preservation*—If $\hat{A}$ and/or $\hat{B}$ are *smaller* (in a well-defined stochastic sense) than $A$ and/or $B$, respectively, then $\hat{X}^s$ is itself *smaller* than $X^s$ and features lower *conditional right-tail expectations* than $X^s$'s.

2. *Variability preservation*—If $\hat{A}$ is *less dispersed* (in a well-defined stochastic sense) than $A$ (for the same $B$) and such dispersion is preserved between $\hat{X}^s$ and $X^s$ (Di Pietro and Sorge 2018a), then $\hat{X}^s$ features lower *conditional right-tail variances* than $X^s$'s.

3. *Variability and skewness comparison*—If $\hat{A}$ and/or $\hat{B}$ are *smaller* than $A$ and/or $B$, respectively, and $X^S$ and $\hat{X}^s$ are both non-negative, then the distribution of $\hat{X}^s$ cannot be *more skewed to the right* (in a well-defined stochastic sense) than that of $X^s$ whenever $\hat{X}^s$ features larger (finite) unconditional variance than $X^s$'s.

How structural properties of a given model change as its constituent components vary is a central question in many areas of economics. By offering new insights on conditional first and higher moments of stationary distributions emerging from linear random recurrences (1), our results are clearly valuable in theoretical explorations of wealth dynamics and within-group inequality, when the recurrence under scrutiny captures the evolution of wealth over time and/or across generations. In doing so, we complement already existing analytical results that solely focus on the determination of Pareto exponents via asymptotic equivalence analysis, e.g., Benhabib et al. (2011). By the same token, our study can easily be exploited to conduct comparative statics exercises in dynamic models featuring Kesten-type reduced-form representations, e.g., Acemoglu and Robinson (2015), Dave and Sorge (2020), and it can provide some discipline for simulation-based validation of theoretical mechanisms proposed in the literature to account for observed higher-order properties of macroeconomic and financial time series, e.g., Gabaix (2009).

In order to illustrate the type of research questions that our study can address, we offer two simple economic applications: the analysis of the role of preference parameters (e.g., bequest intensity) in shaping the long-run distribution of wealth in a stylized dynamic economy with intergenerational wealth transfers and uninsurable risk; and the analysis of the impact of labor earnings taxation on asset accumulation and long-run properties of the wealth distribution in an infinite-horizon economy.

Upon surveying the strands of literature our paper relates to (Section 2), we begin Section 3 with an overview of the pairwise stochastic orderings between random variables (probability distributions), which paves the way to presenting our main results (Section 4). Section 5 lays down two analytically tractable models to frame our comparison theorems in terms of long-run distributional features of wealth and asset accumulation processes. In Section 6, the findings are confronted with existing theoretical studies on the topic. Section 7 offers concluding remarks.

## 2. Literature Review

There are at least two strands of literature our paper speaks to and to which it purports to contribute.

First, linear random recurrences of the form (1) have been extensively used to model the evolution of wealth distributions in the presence of uninsured investment shocks (idiosyncratic returns on financial wealth, embodied in the multiplicative input $A$) and non-diversifiable earnings risk (embodied in the additive input $B$). A burgeoning number of structural dynamic models have been developed that provide micro-foundations for the intergenerational (and/or intertemporal) transmission of wealth, in order to shed light on the sources of wealth inequality across families, the underlying patterns of social mobility and the effects of corrective policies on top wealth concentration.

Benhabib et al. (2011) analyzed the evolution of the wealth distribution in an over-lapping generations (OLG) model with a joy-of-giving bequest motive, in which agents face random labor (additive) and capital (multiplicative) income shocks, which can persist

over time as a function of a latent Markov state. A main finding of their analysis is that, in the presence of uninsurable risk, the upper tail of the stationary wealth distribution is asymptotic to a Pareto law. Benhabib et al. (2015) formally established an analogous *fat upper tail* result for the endogenous distribution of wealth in Bewley economies in which ex-post heterogeneous agents solve an infinite horizon consumption-saving problem with borrowing limits and idiosyncratic shocks to the rates of return on wealth.[2]

Interpreting (1) as a wealth accumulation process in an incomplete market framework and using the so-called Lorenz dominance as an indicator of inequality, Zhu (2013) shows that a mean-preserving rise in the variability of *A* (capital income risk) and/or *B* (earnings risk) entails a less equal stationary distribution of the recurrence (1), provided that it exists. Focusing on the same framework, Peng (2018) jointly studied inequality and mobility dynamics by means of the copula approach, which allows for specifying the joint distribution of percentile ranks of different cross-sectional distributions. In particular, advocating the so-called dispersive order to rank probability distributions in terms of intrinsic variability, it was therein established that inequality is affected by shifts in either type of risk, while mobility reacts only to changes in the capital income one. Di Pietro and Sorge (2018a) singled out a number of inaccuracies in Peng (2018)'s analytical proofs, showing that relatively stronger conditions are needed for the stationary solution of (1) to inherit the stochastic features of the multiplicative input *A*.

In a similar vein, Di Pietro and Sorge (2018b) studied the comparative statics of a well-defined class of wealth transition equations when allowing for stochastically ordered shifts in the multiplicative input. They identified an order-contingent monotone property according to which pure increases in risk foster top wealth concentration, whereas random shifts that involve the average unconditional return process rather lower inequality at the upper end of the stationary distribution. Based on a simple OLG model with intergenerational altruism, Di Pietro and Sorge (2018b) emphasized the potentially ambiguous effects on top wealth inequality of introducing or modifying capital income tax treatments in the presence of non-diversifiable investment risk.

Our paper is also closely related to the ample mathematical work on linear random recurrences, surveyed in, e.g., Buraczewski et al. (2016), which is generally concerned with: (i) the characterization of existence of stationary solutions and the ensuing distributional properties (limit support, unconditional moments, tails); and (ii) the Markov chain properties of stationary solutions such as irreducibility, a-periodicity, mixing, and absolute continuity of the Markov kernel. A small number of papers, such as Müller and Stoyan (2002) and Zhu (2013), have built on this literature by exploring the consequences of distributional shifts in either the multiplicative input *A* or the additive input *B* (or both) on the characteristics of the ensuing stationary distribution for the state variable *X*.

Our study complements the findings from both strands of literature by developing novel comparison theorems that rely on conditional measure of size and variability not explored thus far, theorems that can be applied broadly to any economic or financial model entailing Kesten-type reduced-form dynamics.

## 3. Preliminaries

Consider two continuous, real-valued random variables $Y$ and $\hat{Y}$ with absolutely continuous cumulative distribution functions (CDFs) $\Phi_Y$ and $\Phi_{\hat{Y}}$, respectively. Let $\Phi_Y^{-1}(p) = \inf\{y : \Phi_Y(y) \geq p\}$ and $\Phi_{\hat{Y}}^{-1}(p) = \inf\{y : \Phi_{\hat{Y}}(y) \geq p\}$, $p \in [0,1]$, be the corresponding quantile functions, and

$$S_Y = \left(\lim_{p\downarrow 0}\Phi_Y^{-1}(p),\ \lim_{p\uparrow 1}\Phi_Y^{-1}(p)\right); \qquad S_{\hat{Y}} = \left(\lim_{p\downarrow 0}\Phi_{\hat{Y}}^{-1}(p),\ \lim_{p\uparrow 1}\Phi_{\hat{Y}}^{-1}(p)\right),$$

the supports.[3] Then

**Definition 1.** *$Y$ is said to be smaller than $\hat{Y}$ in the*

(a)   *usual stochastic order – written $Y \leq_{st} \hat{Y}$ – if and only if*

$$\Phi_Y(z) \geq \Phi_{\hat{Y}}(z), \quad \forall z \in (-\infty, \infty);$$

(b)   *dispersive order – written $Y \leq_{disp} \hat{Y}$ – if and only if*

$$\Phi_Y^{-1}(p_2) - \Phi_Y^{-1}(p_1) \leq \Phi_{\hat{Y}}^{-1}(p_2) - \Phi_{\hat{Y}}^{-1}(p_1), \quad \forall (p_1, p_2) : 0 < p_1 \leq p_2 < 1;$$

(c)   *star order – written $Y \leq_\star \hat{Y}$ – if and only if*

$$\frac{\Phi_{\hat{Y}}^{-1}(p)}{\Phi_Y^{-1}(p)} \quad \text{is non-decreasing in } p \in (0,1),$$

*when $Y \geq 0$ and $\hat{Y} \geq 0$ almost surely.*

The *usual stochastic order* is the most common criterion for comparing the location or the magnitude (size) of random variables. It simply states that $Y$ is less likely than $\hat{Y}$ to take on large values, i.e., all values larger than $z$ for any arbitrarily chosen $z$, see e.g., Shaked and Shanthikumar (2007).

The *dispersive order*, among others, can be meaningfully used to compare variability or spread between probability distributions, for it requires that the difference between any two quantiles of $Y$ be smaller than the corresponding quantiles of $\hat{Y}$, see e.g., Lewis and Thompson (1981).

Lastly, the *star order* is a weakening of the well-known Van Zwet (1964)'s convex order, and it has been introduced in the literature to compare the skewness of probability distributions, given its characterization in terms of increasing failure rate on average, see e.g., Oja (1981).[4]

Consider again the linear recurrence (1), with non-negative inputs $(A, B)$. Our main object of interest is the stationary solution to such an equation, defined as follows:

**Definition 2.** *An ergodic, stationary and causal solution $X^s$ to the linear recurrence (1) is a process $\{X_t^s\}$ such that: (i) for all t, $X_t^s$ is a measurable function of $(A_j, B_j)_{j \leq t}$; (ii) $X^s$ is the unique random variable that satisfies $X =^d A_t X + B_t$, where $=^d$ denotes equality in distribution; and (iii) starting from any initial $X_1$, $X_t$ converges in distribution to $X^s$ as $t \to \infty$.*

## 4. Comparison Theorems

Using the stochastic orders defined above, we now present our formal results about pairwise comparisons of stationary solutions to linear random recurrences (1), as enforced by the stochastic inputs $(A, B)$. In the following, $\mathbb{E}[Y]$ and $\mathbb{V}[Y]$ denote the unconditional expectation (mean) and unconditional variance of some real-valued random variable $Y$, respectively; and $\mathbb{E}[Y|\sigma(Z)]$ and $\mathbb{V}[Y|\sigma(Z)]$ denote the conditional expectation and the conditional variance of $Y$ with respect to the $\sigma$-algebra generated by a real-valued random variable $Z$ (both defined on the same probability space), provided they exist, which we assume throughout the paper. To facilitate reading, the proofs of the theorems are collected in Appendix A.

**Theorem 1** (Size preservation). *Consider two linear recurrences*

$$X_{t+1} = A_t X_t + B_t, \quad \hat{X}_{t+1} = \hat{A}_t \hat{X}_t + \hat{B}_t, \qquad t \in \mathbb{Z}^+$$

*where*

i.   *$A_t$, $\hat{A}_t$, $B_t$ and $\hat{B}_t$ are i.i.d. non-negative random variables;*

ii.  *$A_t$ and $B_t$ are statistically independent, $\hat{A}_t$ and $\hat{B}_t$ are statistically independent;*

iii. *$\mathbb{E}[A_t] < 1$, $\mathbb{E}[\hat{A}_t] < 1$, $\mathbb{E}[B_t] < \infty$, $\mathbb{E}[\hat{B}_t] < \infty$;*

iv.   $\hat{A}_t \leq_{st} A_t$ and/or $\hat{B}_t \leq_{st} B_t$ for all $t = 1, 2, \ldots$

Then, there exist stationary solutions $X^s$ under inputs $(A, B)$ and $\hat{X}^s$ under inputs $(\hat{A}, \hat{B})$, with CDFs $\Phi_{X^s}$ and $\Phi_{\hat{X}^s}$ respectively, such that

$$\mathbb{E}\left[X^s \,\middle|\, X^s > \Phi_{X^s}^{-1}(p)\right] \geq \mathbb{E}\left[\hat{X}^s \,\middle|\, \hat{X}^s > \Phi_{\hat{X}^s}^{-1}(p)\right], \tag{2}$$

for all $p \in (0, 1)$.

Theorem 1 stipulates that the stationary solution to (1) inherits the size properties of the inputs $(A, B)$. As a main consequence, any conditional expectation in the right tail (i.e., the conditional mean with respect to the sub-$\sigma$-algebra generated by any quantile of order $p \in (0, 1)$) of the stochastically smaller (in the usual order) stationary distribution never exceeds its counterpart enforced by the stochastically larger input $A$.

**Theorem 2** (Variability preservation). *Consider two linear recurrences*

$$X_{t+1} = A_t X_t + B_t, \qquad \hat{X}_{t+1} = \hat{A}_t \hat{X}_t + B_t, \qquad t \in \mathbb{Z}^+$$

*where*

i.    *$A_t$, $\hat{A}_t$ and $B_t$ are i.i.d. non-negative random variables;*
ii.   *$A_t$ and $B_t$ are statistically independent, $\hat{A}_t$ and $B_t$ are statistically independent;*
iii.  *$\mathbb{E}[A_t] < 1$, $\mathbb{E}[\hat{A}_t] < 1$, $\mathbb{E}[B_t] < \infty$;*
iv.   *$\hat{A}_t \leq_{disp} A_t$ for all $t = 1, 2, \ldots$;*
v.    *$B_t$ has log-concave density.*

Then, there exist stationary solutions $X^s$ under inputs $(A, B)$ and $\hat{X}^s$ under inputs $(\hat{A}, B)$, with CDFs $\Phi_X^s$ and $\Phi_{\hat{X}}^s$, respectively, such that if $\hat{X}^s \leq_{disp} X^s$, it holds

$$\mathbb{V}\left[X^s \,\middle|\, X^s > \Phi_{X^s}^{-1}(p)\right] \geq \mathbb{V}\left[\hat{X}^s \,\middle|\, \hat{X}^s > \Phi_{\hat{X}^s}^{-1}(p)\right], \tag{3}$$

for all $p \in (0, 1)$.

Unlike the usual stochastic order, the dispersive ordering is not invariant under monotone transformations, nor is it generically closed under a product of non-negative random variables. In particular, the dispersive order does not necessarily imply (or is implied by) the usual stochastic order. Theorem 2 asserts that, when conditions for preservation of the dispersive order are fulfilled, e.g., Di Pietro and Sorge (2018a), then any conditional variance in the right tail (i.e., the conditional variance with respect to the sub-$\sigma$-algebra generated by any quantile of order $p \in (0, 1)$) of the less dispersed stationary distribution is bounded from above by its counterpart enforced by the more dispersed input $A$.

**Theorem 3** (Skewnees and variability). *Consider two linear recurrences*

$$X_{t+1} = A_t X_t + B_t, \quad \hat{X}_{t+1} = \hat{A}_t \hat{X}_t + \hat{B}_t, \qquad t \in \mathbb{Z}^+$$

*where*

i.    *$A_t$, $\hat{A}_t$, $B_t$ and $\hat{B}_t$ are i.i.d. non-negative random variables;*
ii.   *$A_t$ and $B_t$ are statistically independent, $\hat{A}_t$ and $\hat{B}_t$ are statistically independent;*
iii.  *$\mathbb{E}[A_t] < 1$, $\mathbb{E}[\hat{A}_t] < 1$, $\mathbb{E}[B_t] < \infty$, $\mathbb{E}[\hat{B}_t] < \infty$;*
iv.   *$\hat{A}_t \leq_{st} A_t$ and/or $\hat{B}_t \leq_{st} B_t$ for all $t = 1, 2, \ldots$*

Then, there exist stationary solutions $X^s$ under inputs $(A, B)$ and $\hat{X}^s$ under inputs $(\hat{A}, \hat{B})$ such that if $X^s \geq 0$ and $\hat{X}^s \geq 0$ almost surely and their first moments are finite, it holds

$$\mathbb{V}\left[\hat{X}^s\right] > \mathbb{V}\left[X^s\right] \quad \Rightarrow \quad \hat{X}^s \nleq_\star X^s. \tag{4}$$

Theorem 3 concerns the relationship between the stochastic magnitude of the multiplicative input $A$, on the one hand, and both the upper tail (conditional) variability and the skewness of the distributions of the stationary solution to (1), on the other. It states that, in the presence of inputs $A$ (and/or $B$) and $\hat{A}$ (and/or $\hat{B}$) that are ranked by their size, and in the case in which the ensuing stationary solutions are non-negative, then the one entailing larger unconditional variance cannot exhibit stronger positive skew than its analogue's.

## 5. Economic Applications

### 5.1. Intergenerational Transfers and Income Risk

Following Zhu (2013), consider an economy populated by a measure-one continuum of agents that live for one period. At the beginning of each period $t$, each agent receives financial bequests $b_t$ from their parents and invests them in a project, whose idiosyncratic (gross) rate of return $R_t$ is an i.i.d. random variable across generations (capital income risk); after $R_t$ realizes and upon obtaining random labor earnings $y_t$, the agent optimally makes their own consumption $c_t \geq 0$ and bequest choices $b_{t+1} \geq 0$ out of their total resources $R_t b_t + y_t$. We assume that $R_t$ and $y_t$ are absolutely continuous, mutually independent, non-negative almost surely and are defined on a bounded support, with $\mathbb{E}[R_t] < 1$ and $\mathbb{E}[y_t] < \infty$. The random variable $y_t$ is also i.i.d. across generations with log-concave density (labor earnings risk). Each agent generates one child at the end of the period, so that the population is stationary over time.

In the case of constant relative risk aversion (CRRA) and additively separable preferences, the agent's problem is

$$\max_{c_t, b_{t+1}} \frac{c_t^{1-\gamma} - 1}{1 - \gamma} + \chi \frac{b_{t+1}^{1-\gamma} - 1}{1 - \gamma}, \tag{5}$$

$$s.t. \quad c_t + b_{t+1} \leq R_t b_t + y_t, \tag{6}$$

where $\gamma > 0$ ($\gamma \neq 1$) is the coefficient of relative risk aversion, and $\chi > 0$ denotes the bequest motive intensity (a measure of intergenerational altruism).[5]

The model admits a unique interior solution

$$c_t = \frac{1}{1 + \chi^{-\frac{1}{\gamma}}} (R_t b_t + y_t), \quad b_{t+1} = \frac{\chi^{-\frac{1}{\gamma}}}{1 + \chi^{-\frac{1}{\gamma}}} (R_t b_t + y_t), \tag{7}$$

hence the (financial) wealth accumulation process reads as

$$\omega_{t+1} = \frac{1}{1 + \chi^{-\frac{1}{\gamma}}} R_t \omega_t + \frac{1}{1 + \chi^{-\frac{1}{\gamma}}} y_t, \tag{8}$$

where $\omega_t \equiv b_t$. Apparently, Equation (8) is in the same form as Equation (1) with $\omega_t = X_t$, $A_t = \left(1 + \chi^{-\frac{1}{\gamma}}\right)^{-1} R_t$ and $B_t = \left(1 + \chi^{-\frac{1}{\gamma}}\right)^{-1} y_t$.

Consider now two distinct economies, $E$ and $\hat{E}$, the former being characterized by a stronger intensity of the bequest motive: $\chi_E > \chi_{\hat{E}}$. For a given $\gamma > 0$, we therefore have $\left(1 + \chi_E^{-\frac{1}{\gamma}}\right)^{-1} > \left(1 + \chi_{\hat{E}}^{-\frac{1}{\gamma}}\right)^{-1}$, which in turn implies $\hat{A}_t \leq_{st} A_t$ and $\hat{B}_t \leq_{st} B_t$. By Theorem 3.B.4 in Shaked and Shanthikumar (2007), we also have $\hat{A}_t \leq_{disp} A_t$ (and $\hat{B}_t \leq_{disp} B_t$). Note that all the preliminary conditions of Theorems 1 and 2 are met; hence, stationary solutions $\omega^s$ and $\hat{\omega}^s$ in the two economies exist and can be ranked by the usual stochastic order. We can therefore state the following.

**Proposition 1.** *The stationary distribution of wealth in economy E features higher conditional expected concentration in the right tail than economy Ê's. If, in addition, $\omega^s \geq_{disp} \hat{\omega}^s$, then the former also displays a larger right-tail conditional variability than the latter.*

　　Intuitively, the same logic as the one underlying Proposition 1 can be used to study the effects on conditional tail first and second moments of the stationary distribution of wealth of (i) changes in the degree of risk aversion of agents, and/or (ii) fiscal policies that affect the intergenerational transmission of wealth (e.g., bequest taxation).

*5.2. Dynamic Asset Accumulation and Earnings Taxation*

　　Following Acemoglu and Robinson (2015) and Di Pietro and Sorge (2018b), consider an economy populated by a measure-one continuum of infinitely lived individuals, indexed by $i$, with heterogeneous asset holdings. Assume that each individual $i$ consumes a constant fraction $\theta \in (0,1)$ of their wealth (cash on hand) $w_{i,t}$ at any time $t$, while facing an i.i.d. labor earnings process $(y_{i,t})$—with $E[y_{i,t}] \in (0, \infty)$ and finite variance—and a random i.i.d. sequence of net rates of return $(r_{i,t})$ with CDF $\Phi_r$, $\lim_{p \downarrow 0} \Phi_r^{-1}(p) = -1$ and $E[r_{i,t}] = 0$.

　　Letting $\tau \in (0,1)$ be a flat-rate tax on labor earnings, the asset dynamic evolution for individual $i$ is given by

$$w_{i,t+1} = (1 + r_{i,t} - \theta)w_{i,t} + (1 - \tau)y_{i,t}, \tag{9}$$

which again is in the same form as (1) with $w_{i,t} = X_t$, $A_t = (1 - \theta) + r_{i,t}$ and $B_t = (1 - \tau)y_{i,t}$.

　　Let us now consider two different earnings tax rates, $\tau$ and $\hat{\tau}$, satisfying $0 < \tau < \hat{\tau} < 1$, from which it readily follows that $(1 - \tau)B_t \geq_{st} (1 - \hat{\tau})B_t$. Since all the preliminary conditions of Theorem 1 are met, we obtain the following

**Proposition 2.** *In the dynamic asset accumulation model, average wealth concentration for any right tail quantile of the stationary distribution decreases with the earnings tax rate.*

　　Clearly, the logic of Proposition 2 is fully retained in all cases where a shift in the distribution of net rates of return $r_{i,t}$ and/or of labor earnings $y_{i,t}$ is obtained, which complies with the usual stochastic order, e.g., the introduction of a uniform tax on capital income, a behavioral change in the marginal propensity to saving, or a shift in investment average risk, each inducing rates of returns $\hat{r}_{i,t}$ satisfying $r_{i,t} \geq_{st} \hat{r}_{i,t}$. For the case of capital income taxation, this observation allows us to state the following[6]

**Corollary 1.** *Consider the dynamic asset accumulation model, and let $\varphi \in (0,1)$ denote the tax levied on returns $r_{i,t}$. Then, average wealth concentration for any right tail quantile of the stationary distribution decreases with the tax rate $\varphi$.*

　　By the same token, since for non-negative random variables $r_{i,t}$ and $\hat{r}_{i,t}$ one can have $r_{i,t} \geq_{st} \hat{r}_{i,t}$ and $\mathbb{V}[r_{i,t}] < \mathbb{V}[\hat{r}_{i,t}]$, the content of Theorem 3 can also become operative: in such a case, a larger unconditional variability of the stationary distribution of wealth induced by the stochastically smaller rates of return $\hat{r}_{i,t}$ relative to the distribution arising under $r_{i,t}$ would imply that the former displays no smaller positive skew than the latter.[7]

## 6. Discussion

　　As stressed in our literature review, extant studies on the topic of stochastic comparisons of probability distributions have established a number of analytical results concerning either *Pareto exponents* as an indicator for top wealth inequality, e.g., Benhabib et al. (2011, 2015), Di Pietro and Sorge (2018b), or *unconditional* measures of spread (dispersion), when framed in the context of dynamic wealth accumulation processes that exhibit ergodic behavior, e.g., Zhu (2013), Peng (2018), Di Pietro and Sorge (2018a). In the present paper, we formalize stochastic comparisons in terms of *conditional* first and second moments as well as positive skew of stationary distributions that are not limited to the asymptotic properties of the counter-cumulative distribution function, for they readily apply to *any given quantile* that a researcher might be interested in, e.g., identifying the middle and upper class of a given socio-economic system, or even the whole distribution except for the lowest region of its support.

In terms of studies of the long-run effects of earnings taxation in incomplete market economies, a key result in Benhabib et al. (2011) implies that the Pareto exponent—which is inversely related to top wealth concentration—would be invariant with respect to the implementation of such a fiscal policy, under conditions that warrant existence of a power–law approximation of the right tail of the stationary distribution. Our analysis (Proposition 2) complements Benhabib et al. (2011)'s result in terms of right conditional expectations, even in cases where the stationary distribution exhibits no fat-tailed behavior, showing that uniform earnings taxation affects expected concentration of wealth in higher (even extreme) quantiles. A corollary of this result is that capital income and earnings taxes produce qualitatively similar effects in terms of conditional tail expectations: the higher the tax rate, the lower the conditional measure of size for the underlying stationary distribution, whatever the fiscal policy in place.

In the same vein, provided that the mild sufficient conditions for the preservation of the dispersive order developed in Di Pietro and Sorge (2018a) are met, Theorem 2 can be advocated to claim that the conditional right tail variability of wealth increases with the tax rate on labor earnings; this suggests that top wealth inequality, as measured by the dispersion of wealth holdings in extreme right tail quantiles, *can be* affected by the earnings tax regime, while the tail index (if it exists) offers no insights on this (being solely driven by the properties of the multiplicative input $A$), as shown in Benhabib et al. (2011).

## 7. Concluding Remarks

A natural question concerns the generalizability of the present analysis to the multi-variate case, as well as its pragmatic value when it comes to evaluating size and variability features of empirical stationary distributions. On the first point, the advanced tools for the analysis of multivariate linear recurrences surveyed in Buraczewski et al. (2016) can be brought to bear upon the problem of the existence of a strictly stationary solution and its conditional tail behavior in multivariate systems. On the second point, being framed in terms of conditional moments of stationary solutions to linear random recurrences, our theoretical results are not tied to any specific model formalizing economic behavior, and they therefore cover a broad scope of applications. For instance, they can be used to quantify the differential effects of capital income versus earnings taxation in fully fledged life-cycle models calibrated to match the salient features of the empirical income and wealth distributions in a given country, as in Benhabib et al. (2011). We leave these aspects for future work.

**Author Contributions:** Conceptualization, M.M.S.; formal analysis, C.D.P., M.P. and M.M.S.; writing—original draft preparation, M.M.S. All authors have read and agreed to the published version of the manuscript.

**Funding:** This research received no external funding.

**Informed Consent Statement:** Not applicable.

**Conflicts of Interest:** The authors declare no conflict of interest.

## Appendix A. Proofs

*Appendix A.1. Proof of Theorem 1*

We first show that, for any given non-negative input $(A, B)$ complying with the stated assumptions, a unique stationary solution to the linear recurrence (1) exists. By Jensen's inequality, we have $\mathbb{E}[\log A_t] \leq \log \mathbb{E}[A_t] < 0$ and $\mathbb{E}[\log^+ B_t] \leq \mathbb{E}[B_t] < \infty$, where $\log^+ B_t := \max\{0, \log B_t\}$. Note that the same holds for the inputs $(\hat{A}, \hat{B})$. These conditions are sufficient for existence of the ergodic, causal stationary solutions $X^s$ and $\hat{X}^s$ to the linear recurrences introduced in Theorem 1, see, e.g., Müller and Stoyan (2002).

Pick now $X_1 = \hat{X}_1$. Since $X_1$ ($\hat{X}_1$) and $A_1$ ($\hat{A}_1$) are mutually independent, and the latter is restricted to be non-negative, then the bi-variate function $\psi_1(A_1, X_1) = A_1 \cdot X_1$ ($\psi_1(\hat{A}_1, \hat{X}_1) : \hat{A}_1 \cdot \hat{X}_1$) is non-decreasing.[8] By Theorem 1.A.3.b in Shaked and Shanthikumar

(2007), it follows that $\hat{A}_1 \hat{X}_1 \leq_{st} A_1 X_1$, and since the function $\psi_2(A_1 X_1, B_1) = A_1 X_1 + B_1$ (the function $\psi_2(\hat{A}_1 \hat{X}_1, \hat{B}_1) = \hat{A}_1 \hat{X}_1 + \hat{B}_1$) is itself non-decreasing and $B_1$ ($\hat{B}_1$) is assumed to be independent of both $A_1$ ($\hat{A}_1$) and $X_1$, by repeated application of Theorem 1.A.3.b in Shaked and Shanthikumar (2007), we obtain $\hat{X}_t \leq_{st} X_t$ for all $t \geq 2$. Note that, by Definition 2, the sequence $\{X_j, j = 1, 2, \}$ converges in distribution to $X^s$, and the sequence $\{\hat{X}_j, j = 1, 2, \}$ converges in distribution to $\hat{X}^s$ as $j \to \infty$. By Theorem 1.A.3.c in Shaked and Shanthikumar (2007), it follows that $\hat{X}^s \leq_{st} X^s$.

It is well known that $\hat{X}^s \leq_{st} X^s$ if and only if $\mathbb{E}[g(\hat{X}^s)] \leq_{st} \mathbb{E}[g(X^s)]$ for all non-decreasing functions $g(\cdot)$ for which the expectations exist. Thus, $\hat{X}^s \leq_{st} X^s$ implies $\mathbb{E}[h(\hat{X}^s)] \leq_{st} \mathbb{E}[h(X^s)]$ for all non-decreasing and convex functions $h$, provided that the expectations exist. This weak inequality characterizes the increasing convex order—see Definition 4.A.1 in Shaked and Shanthikumar (2007)—which in turn is equivalent to the following

$$\int_p^1 \Phi_{\hat{X}^s}^{-1}(u) du \leq \int_p^1 \Phi_{X^s}^{-1}(u) du, \quad \forall p \in (0, 1), \tag{A1}$$

by virtue of Theorem 4.A.3 in Shaked and Shanthikumar (2007). The assertion then follows immediately.

*Appendix A.2. Proof of Theorem 2*

By virtue of Theorem 1, the stationary solutions $X^s$ and $\hat{X}^s$ exist and satisfy $\hat{X}^s \leq_{st} X^s$. If these are also ranked by the dispersive order, i.e., $\hat{X}^s \leq_{disp} X^s$, then it holds

$$\mathbb{E}\left[\varphi\left(\hat{X}^s - \mathbb{E}[\hat{X}_p^s]\right) \Big| \hat{X}^s > \Phi_{\hat{X}^s}^{-1}(p)\right] \leq \mathbb{E}\left[\varphi\left(X^s - \mathbb{E}[X_p^s]\right) \Big| X^s > \Phi_{X^s}^{-1}(p)\right], \quad \forall p \in (0, 1), \tag{A2}$$

for all convex real-valued functions $\varphi$, where

$$\mathbb{E}\left[\hat{X}_p^s\right] = \frac{1}{1-p} \int_p^1 \Phi_{\hat{X}^s}^{-1}(u) du, \quad \mathbb{E}\left[X_p^s\right] = \frac{1}{1-p} \int_p^1 \Phi_{X^s}^{-1}(u) du, \quad 0 < p < 1, \tag{A3}$$

provided the measures in (A2) exist, by Theorem 4 in Sordo (2009), and the fact that the dispersive order implies the right spread (also known as excess wealth) order for random variables with finite means. Taking $\varphi$ to be the squaring function delivers the assertion—see also Corollary 5 in Sordo (2009).

*Appendix A.3. Proof of Theorem 3*

By virtue of Theorem 1, the stationary solutions $X^s$ and $\hat{X}^s$ exist and satisfy $\hat{X}^s \leq_{st} X^s$. Let both be non-negative, and assume their first and second (central) moments are finite, with $\mathbb{V}[\hat{X}^s] > \mathbb{V}[X^s]$. Then, by virtue of Theorem 3.B.16 in Shaked and Shanthikumar (2007), $\hat{X}^s$ cannot be less dispersed then $X^s$, and thus it cannot be smaller than $X^s$ in the star order; see Bartoszewicz (1985).

## Notes

[1]  Stochastic recurrence equations (1) are also known as *random coefficient autoregressive models* (Regis et al. 2022), *Kesten equations* (Buraczewski et al. 2016), *linear recursions with multiplicative and additive noise*, see e.g., Benhabib and Dave (2014); Dave and Sorge (2020, 2021).

[2]  See Gabaix (2009) and Benhabib and Bisin (2018) for excellent surveys on Kesten-type mechanisms for generating power-law behavior in numerous contexts of interest.

[3]  The inverse of a monotone function (which is not strictly monotone) is taken to be the right continuous version of it.

[4]  The star order is known to imply the so-called Lorenz dominance provided the compared random variables exhibit equal unconditional means. Lorenz dominance has been widely used in the economic literature as a partial ordering criterion for comparisons across income or wealth distributions, see e.g., Zhu (2013).

[5]  CRRA preferences are widespread in theoretical studies of wealth distributions, for they induce linear decision rules, see e.g., Benhabib et al. (2011); Zhu (2013).

[6]  Since $(1 - \varphi) r_{i,t} \leq_{st} r_{i,t}$ for all constant $\varphi \in (0, 1)$, the assertion follows immediately from Theorem 1.

[7]   Consider e.g., $r_{i,t}$ to be uniformly distributed over the $[1, 2]$ interval, and $\hat{r}_{i,t}$ to be uniformly distributed over the $[0, 2]$ interval. Apparently, $\hat{r}_{i,t} \leq_{st} r_{i,t}$ and $\mathbb{V}[r_{i,t}] < \mathbb{V}[\hat{r}_{i,t}]$.

[8]   We remark that throughout the book by Shaked and Shanthikumar (2007), increasing means weakly increasing, i.e., non-decreasing.

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
