# Peer review of "Stochastic Ordering of Stationary Distributions of Linear Recurrences: Further Results and Economic Applications"

_economies, doi:10.3390/economies11040125_

Round 1

Reviewer 1 Report

Since the paper is a theoretical study, deriving new properties on already established constructs, I recommend expanding the literature review section (currently included in the introduction) with more studies (probably of applied nature since the seminal works were already cited) and adding a discussion section where the findings are confronted with existing theoretical and practical studies on the topic.

Have the authors considered performing a Monte Carlo-type experiment to illustrate their findings on data?

Author Response

We wish to thank the Reviewer for his/her careful reading of the paper and for his/her many constructive comments and suggestions. All of these concerns have been thoroughly discussed among the co-authors and, as a result, we believe that a much improved version of our paper has been obtained in the revision.

Following the Reviewer's advice, in the revised version of our manuscript we have expanded the literature review section, separating it from the Introduction, and added a discussion section (section 6) where our analytical findings are confronted with existing theoretical studies on the topic. 

In particular, we emphasize how existing studies on stochastic comparisons of probability distributions have established a number of analytical results concerning either Pareto exponents as an indicator for top wealth inequality (Benhabib et al., 2011, 2015; Di Pietro and Sorge, 2018b), or unconditional measures of spread (dispersion), when framed in the context of dynamic wealth accumulation processes that exhibit ergodic behavior (Zhu, 2013; Peng, 2018; Di Pietro and Sorge, 2018a). In the present paper, we formalize stochastic comparisons in terms of conditional first and second moments as well as positive skew of stationary distributions, that are not limited to the asymptotic properties of the counter-cumulative distribution function, for they readily apply to any given quantile a researcher might be interested in, e.g. that identifying the middle and upper class of a given socio-economic system, or even the whole distribution except the lowest region of its support.

In terms of studies of the long-run effects of earnings taxation in incomplete market economies, a key result in Benhabib et al. (2011) implies that the Pareto exponent – which is inversely related to top wealth concentration – would be invariant with respect to the implementation of such a fiscal policy, under conditions that warrant existence of a power-law approximation of the right tail of the stationary distribution. Our analysis (Proposition 2) qualifies Benhabib et al. (2011)’s result in terms of right conditional expectations, even in cases where the stationary distribution exhibits no fat-tailed behavior, showing that uniform earnings taxation does affect expected concentration of wealth in higher (possibly extreme) quantiles. A corollary of this result is that capital income and earnings taxes produce qualitatively similar effects in terms of conditional tail expectations: the higher the tax rate, the lower the conditional measure of size for the underlying stationary distribution, whatever the fiscal policy in place.

A natural question concerns the generalizability of the present analysis to the multivariate case, as well as its pragmatic value when it comes to evaluating size and variability features of empirical stationary distributions. On the first point, the advanced tools for the analysis of multivariate linear recurrences surveyed in Buraczewski et al. (2016) can be brought to bear upon the problem of the existence of a strictly stationary solution and their conditional tail behavior in multivariate systems. On the second point, it should be remarked that, since framed in the context of conditional moments of stationary solutions to linear random recurrences, our theoretical results are not tied to any specific model formalizing economic behavior, and therefore cover a broad scope of applications. For instance, they can be used to quantify the differential effects of capital income versus earnings taxation, in fully-fledged lyfe-cycle models calibrated to match the salient features of the empirical income and wealth distributions in a given country, as in Benhabib et al. (2011). We leave these aspects for future work.

Reviewer 2 Report

The manuscript is well-written and organized. The scientific contribution could be enhanced by adding more results and analysis but this is not mandatory. I suggest to publish the manuscript as is.

Regards

Author Response

We wish to thank the Reviewer for his/her careful reading of the paper and for his/her many constructive comments and suggestions. All of these concerns have been thoroughly discussed among the co-authors and, as a result, we believe that a much improved version of our paper has been obtained in the revision.

Following the Reviewer's advice, in the revised version of our manuscript we have spelled out a number of implications of our analysis by adding new results (on e.g. the implications of capital income taxation for the determination of the conditional measures of size and variability studied in the paper, see Corollary 1) and also linking them to extant findings in the literature that rather focus exclusively on Pareto exponents as the relevant measure of top wealth inequality (see the brand new section Discussion).

Reviewer 3 Report

The content of the paper is interesting. The study subject is relevant. The presentation of the result is acceptable. Analysis of the state of the problem requires improvement. The abstract should be modified taking into account the goal of the study and the proposed principal result.

Author Response

We wish to thank the Reviewer for his/her careful reading of the paper and for his/her many constructive comments and suggestions. All of these concerns have been thoroughly discussed among the co-authors and, as a result, we believe that a much improved version of our paper has been obtained in the revision.

Following the Reviewer's advice, in the revised version of our manuscript we have improved the presentation of the state-of-the-art literature, the problem statement and the discussion of the scope of our results, by adding new results (on e.g. the implications of capital income taxation for the determination of the conditional measures of size and variability studied in the paper) and also linking them to extant findings in the literature that rather focus exclusively on Pareto exponents as the relevant measure of top wealth inequality, see the brand new section Discussion. The Abstract has also been modified to reflect the Reviewer's suggestion.